# *Let's Synthesize Step by Step*: Iterative Dataset Synthesis with Large Language Models by Extrapolating Errors from Small Models

**Ruida Wang**[*][H]    **Wangchunshu Zhou** [A]    **Mrinmaya Sachan** [E]

[H] HKUST          [A] AIWaves Inc.          [E] ETH Zürich

rwangbr@connect.ust.hk chunshu@aiwaves.cn msachan@ethz.ch

## Abstract

*Data Synthesis* is a promising way to train a small model with very little labeled data. One approach for data synthesis is to leverage the rich knowledge from large language models to synthesize pseudo training examples for small models, making it possible to achieve both data and compute efficiency at the same time. However, a key challenge in data synthesis is that the synthesized dataset often suffers from a large distributional discrepancy from the *real task* data distribution. Thus, in this paper, we propose *Synthesis Step by Step* (**S3**), a data synthesis framework that shrinks this distribution gap by iteratively extrapolating the errors made by a small model trained on the synthesized dataset on a small real-world validation dataset using a large language model. Extensive experiments on multiple NLP tasks show that our approach improves the performance of a small model by reducing the gap between the synthetic dataset and the real data, resulting in significant improvement compared to several baselines: 9.48% improvement compared to ZeroGen, 2.73% compared to GoldGen, and 15.17% improvement compared to the small model trained on human-annotated data.[1]

## 1 Introduction

Large Language Models (LLMs) (Brown et al., 2020; Chowdhery et al., 2022; Touvron et al., 2023; OpenAI, 2023) have shown promising zero-shot performance on a wide range of tasks, demonstrating their potential of serving as generalist models. However, LLMs suffer from efficiency issues due to large model sizes and high inference latency, making them hard to deploy in real-world applications. Therefore, small models trained on task-specific data are still favored in many resource-constrained scenarios because they have much

---

*[*] Work done while at exchange at ETH Zürich

[1] The code and generated data can be found at https://github.com/RickySkywalker/Synthesis_Step-by-Step_Official

fewer parameters, are easy to deploy, and perform well in specific downstream tasks (Xu et al., 2021).

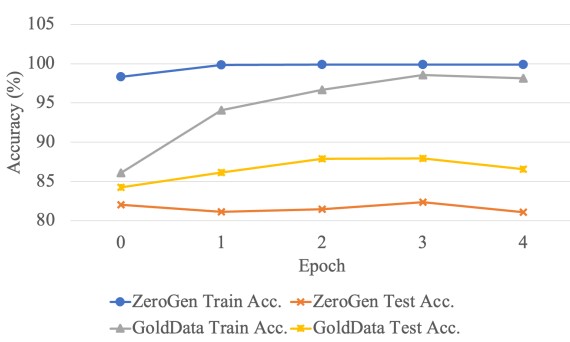

Figure 1: Training and testing accuracy of DistilBert with ZeroGen (Ye et al., 2022b) on the IMDb dataset with 200k training datapoints. Also shown are the training and testing accuracy of the model trained on Gold-Data. We can see here that ZeroGen's training accuracy quickly reaches nearly 100%, but testing accuracy remains low.

However, fitting a small model for a specific task may require large amounts of human-labeled data, which is not available in many downstream tasks and is expensive to annotate. This data inefficiency problem makes it challenging to fine-tune a small model. Therefore, a number of distinct research approaches attempt to reduce the amount of data required for fine-tuning small models on specific tasks, including knowledge distillation (Hinton et al., 2015; Beyer et al., 2022; Hsieh et al., 2023; Xu et al., 2020; Zhou et al., 2020; Shridhar et al., 2023), data augmentation (DeVries and Taylor, 2017; Shorten and Khoshgoftaar, 2019; Li et al., 2022), module replacing (Xu et al., 2020; Zhou et al., 2023), semi-supervised learning (Chen et al., 2020; Wang et al., 2021; Smith et al., 2022), and data synthesis (Anaby-Tavor et al., 2020; Puri et al., 2020).

In this work, we focus on data synthesis, which generates data and corresponding labels from scratch. Unlike semi-supervised learning, which

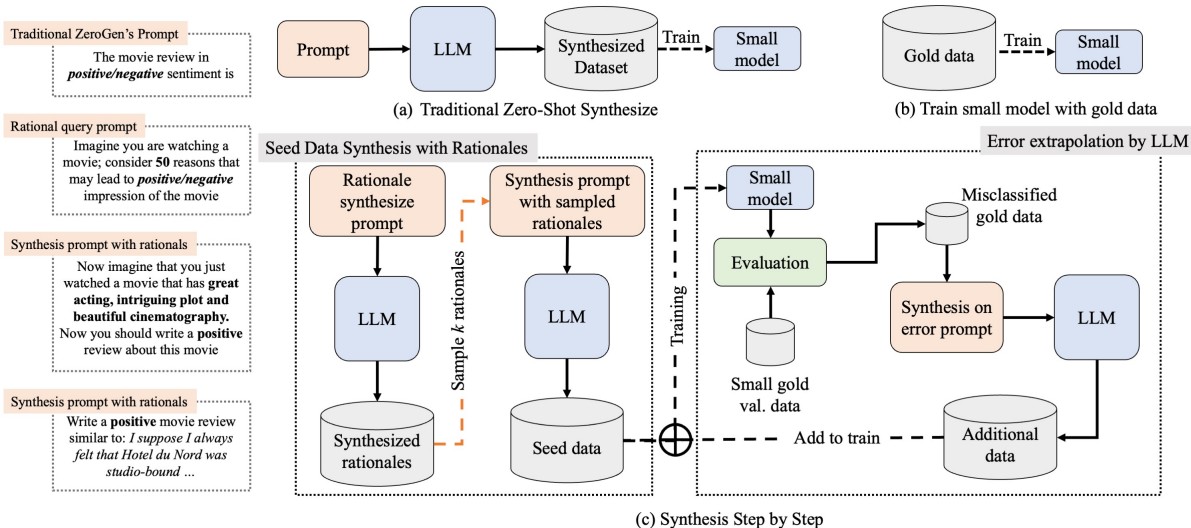

Figure 2: Both (a) traditional zero-shot dataset synthesis methods and (b) training small models directly on gold data do not leverage feedback from the small model trained on the synthesized dataset. In contrast, (c) our approach, S3, first synthesizes a seed dataset in a zero-shot fashion with rationales (left-hand side). Then, we iteratively reduce the gap between the synthesized data distribution and the gold data distribution by extrapolating the errors of a small model trained on the currently synthesized data on a small gold validation set. The additional synthesized data can, therefore, be considered to be sampled from the difference between the currently synthesized data distribution and gold data distribution. By mixing it with the currently synthesized data, we can recover the gold data distribution and therefore improve the performance of a small model trained on the data mixture.

relies on unlabeled data, this approach is simpler and more efficient, especially when unlabeled data is scarce. Most existing methods in data synthesis for NLP utilize LLMs to generate an unlimited amount of training data for training a small model.

Existing dataset synthesis methods typically require a massive amount of synthesized data to achieve relatively good performance with a small model, like in ZeroGen (Ye et al., 2022b), which sometimes needs as much as 1M records of synthesized data. However, this often results in additional data synthesis cost and computation costs when training the small task-specific model.

Intuitively, the quality of the synthesized data, or the extent to which the synthesized data resembles the gold task data, is crucial for the small model's performance. However, due to the complexity of specific tasks in the real world, the synthesized data often suffers from a distribution gap from the real-world data distribution. This can be clearly seen in Fig.1. The small model's training accuracy on synthesized data is close to 100% but the testing accuracy on real-world data is still low. In contrast, the gap between training and testing accuracy is much smaller when trained on human-annotated data.

To reduce the distribution gap and improve data efficiency in dataset synthesis, we propose *Synthesis Step by Step* (**S3**), a novel dataset synthesis framework that reduces the distribution gap in a data-efficient way by dynamically optimizing the synthesized dataset. As illustrated in Fig. 2, S3 first synthesizes a *seed dataset* with an explain-then-generate method that first prompts LLMs to generate rationales for each label and then combines the generated rationale and task-specific prompts to generate data points. S3 then refines the seed dataset by iteratively synthesizing more data by extrapolating the errors of a model trained on the seed dataset made on a small validation set, which we assume is sampled from the real task data distribution.

We summarize our contribution as follows: (1) We propose a novel point of view for dynamic dataset synthesis, which allows for the creation of training data for smaller models and can be optimized by adding more data; based on this point of view, we propose the S3 framework that can synthesize and optimize a pseudo dataset using LLM that can efficiently shrink the distribution gap in dataset synthesis. (2) We perform a theoretical analysis for the effectiveness of S3 on reducing the distribution gap. (3) We perform extensive experiments on three major NLP tasks and obtain

an average 9.48% improvement compared to Ze-roGen (Ye et al., 2022b), a representative baseline for dataset synthesis, using only 30.43% of data on average.

## 2 Methodology

We describe the proposed S3 framework in detail in this section. The key idea of S3 is to first synthesize a *seed dataset* by prompting LLMs and then to iteratively reduce the distribution gap by extrapolating errors the small model makes on a small validation set from the gold data distribution. S3 comprises the following steps:

1. **Seed data generation:** We utilize an LLM to analyze the task we are working on, then synthesize a list of possible rationales for such a task. If the task is hard to analyze, we can skip this step. Then, we combine the synthesized rationales, possible context sentences, and labels in one prompt to guide the LLM to synthesize the dataset.

2. **Small model training:** Train the small model with the synthesized dataset, then validate the small model on real-world validation data, and attain misclassified data of the small model, use them as errors.

3. **Error extrapolation:** Use the LLM to extrapolate the errors of the small model and synthesize additional data using the information in errors.

4. **Combine and Repeat:** Combine the additional dataset and original dataset as a new synthesized train dataset for the small model, then repeat steps 2 and 3 for multiple rounds until the performance of the small model converges.

We first introduce some background and key notations in Section 2.1. We then describe the algorithms for seed data synthesis and iterative error extrapolation-based synthesis in Section 2.2 (point 1. above) and Section 2.3 (points 2, 3, 4 above), respectively. Finally, we give a theoretical interpretation of the proposed method in Section 2.6.

### 2.1 Background

Following Sharp et al. (2017), we denote the distribution of human language for the LLM under prompt input $\mathcal{T}$ as $\mathbb{P}_{LLM}(\cdot|\mathcal{T})$. The *small model* is a computationally efficient model that will be trained on our synthesized dataset. In general, the small model contains much fewer parameters and is easy to train and deploy in real-world applications. We denote a small model trained by dataset $\mathcal{D}_{train}$ as $f(\cdot|\mathcal{D}_{train})$.

### 2.2 Seed Data Synthesis with Rationales

*Seed Data* is defined as the basic zero-shot synthesized dataset for our S3 framework.

---

**Algorithm 1:** Seed data synthesis with rationales

**Input:** $\mathcal{Y}, \mathcal{T}_{ration}, \mathcal{T}_{query}^{(1)}, \mathbb{P}_{LLM}, K, k, N_{seed}$
**Output:** $\mathcal{D}_{seed}$

1 **for** *each $y_i \in \mathcal{Y}$* **do**
2 $\quad \lfloor \; r_i \leftarrow topK(\mathbb{P}_{LLM}(\cdot|\mathcal{T}_{ration}(y_i))$
3 $\mathcal{D}_{seed} \leftarrow \emptyset$
4 **for** *i in range($N_{seed}$)* **do**
5 $\quad y_{curr} \sim U_1(\mathcal{Y})$
6 $\quad r_{curr} \sim U_k(r_i)$
7 $\quad x_{curr} \sim \mathbb{P}_{LLM}(\cdot|\mathcal{T}_{query}^{(1)}(r_{curr}, y_{curr}))$
8 $\quad \mathcal{D}_{seed} \leftarrow \mathcal{D}_{seed} \cup \{(x_{curr}, y_{curr}\}$

---

We present the algorithm for seed data synthesis with rationales in Alg. 1. Here, $\mathcal{Y}$ denotes the set of all possible labels in the task we are working on; $\mathcal{T}_{ration}(y)$ denotes label and task descriptive prompt for rationales synthesis; $\mathcal{T}_{query}^{(1)}(r, y)$ is the data synthesis prompt that wraps the rationales in $r$ and the label $y$ together to query LLM for a data point; $topK$ means top-K sampling from the LLM outputs to obtain the rationale list for a specific label; $U_i(S)$ means uniformly sample $i$ non-repeating elements in set $S$. The resulting seed dataset is denoted as $\mathcal{D}_{seed} = \{\mathcal{X}_{seed}, \mathcal{Y}_{seed}\}$.

For instance, for the IMDb (Maas et al., 2011) dataset, a sentiment analysis dataset on movie reviews, $\mathcal{T}_{ration}(y_i = positive/negative)$ is: "*What is the reason that may lead to a **positive/negative** movie review.*" and the $\mathcal{T}_{query}(r_{curr}, positive)$ is: "*Now imagine that you just watched a movie that has **great acting**, **intriguing plot**, and **beautiful cinematography**. Now you should write a **positive** review about this movie.*" We use the prompt as an input to the LLM and obtain the target output as the synthesized pseudo example. This "explain-then-generate" approach enables us to generate more diverse, informative, and realistic examples.

## 2.3 Dataset Refinement with Error Extrapolation

We then describe the *Error Extrapolation-based Synthesis* (EES) framework that attempts to iteratively reduce the distribution gap by extrapolating the errors of a small model trained on the currently synthesized dataset on a small validation set. This is different from conventional data synthesis methods, where the synthesized dataset is fixed after finishing the synthesis process and is used for training the small model. Specifically, the EES process extrapolates errors made by small models on the real-world validation datasets to synthesize some additional data to fix the error.

We use two different data sources in the EES process: the seed dataset ($\mathcal{D}_{seed}$), and a small human-labeled, real-world dataset referred to as *gold data*, denoted as $\mathcal{D}_{gold}$. In EES, we first divide the gold data into a validation dataset $\mathcal{D}_{gold}^{(val)}$ and a testing dataset $\mathcal{D}_{gold}^{(test)}$. We use $\mathcal{D}_{gold}^{(val)}$ to find and fix the distribution gap and use $\mathcal{D}_{gold}^{(test)}$ to judge the performance of the small model.

---

**Algorithm 2:** Algorithm for Error Extrapolation

**Input:** $\mathcal{D}_{seed}, \mathcal{D}_{gold}^{(eval)}, \mathcal{D}_{gold}^{(test)}, f, \mathbb{P}_{LLM}, R, \mathcal{T}_{mis}^{(1)}$
**Output:** $\mathcal{D}_{train}$

1   $\mathcal{D}_{add}^{(0)} \leftarrow \emptyset$
2   **for** $q$ *in range($R$)* **do**
3     $init(f)$; // reinitialize $f$ (clear last round's train)
4     $\mathcal{D}_{train}^{(q)} \leftarrow \mathcal{D}_{seed} \cup (\cup_{i=1}^{q} \mathcal{D}_{add}^{(i)})$
5     $train(f, \mathcal{D}_{train}^{(q)})$
6     $\mathcal{D}_{mis}^{(q)} \leftarrow misclass\{f(\mathcal{D}_{gold}^{(eval)}|\mathcal{D}_{train}^{(q)})\}$
7     $\mathcal{D}_{add}^{(q+1)} \leftarrow \emptyset$
8     **for** *each* $(x_{mis}, y_{mis}) \in \mathcal{D}_{mis}^{(q)}$ **do**
9       $x_{add} \sim \mathbb{P}_{LLM}(\cdot | \mathcal{T}_{mis}^{(1)}(x_{mis}, y_{mis}))$
10      $\mathcal{D}_{add}^{(q+1)} \leftarrow \mathcal{D}_{add}^{(q+1)} \cup \{(x_{add}, y_{mis})\}$
11   $\mathcal{D}_{train} \leftarrow \mathcal{D}_{seed} \cup (\cup_{i=1}^{N} \mathcal{D}_{add}^{(i)})$

---

We present the whole process of EES in Alg. 2. One round in the for-loop beginning at line 2 denotes one round of EES. $R$ denotes the number of rounds of EES we want to perform; in our implementation, we typically do 2 rounds of experiments. $f$ denotes the small model; $\mathcal{D}_{mis}^{(q)}$ denotes the set of examples mis-classified by the small model on the gold validation dataset in the $q$-th round of EES. $\mathcal{T}_{mis}^{(1)}(x_{mis}, y_{mis})$ denotes the prompt used for error extrapolation. The prompt asks the LLM to synthesize a data point similar to $x_{mis}$ with label $y_{mis}$. In our implementation, we use the prompt: "*Write a **positive** movie review like **The movie is great**.*" $\mathcal{D}_{add}^{(q+1)}$ denotes the $q + 1$-th additional dataset we synthesized on LLM based on extrapolating $\mathcal{D}_{mis}^{(q)}$.

The key steps of the EES algorithm are to train the small model with the current synthesized dataset (line 6) and utilize the LLM to extrapolate the misclassified data to generate more training data (lines 8-10). This creates a dataset that better reflects the underlying truth.

In sum, the EES process reduces the distribution gap by using the misclassified data to model the distribution gap and using the LLM to sample additional data points from it. This idea is similar to doing optimization on the residuals in the gradient boosting literature (Friedman, 2002).

## 2.4 Special process for multi-sentence task

For clarity, we focus on single-sentence tasks in our algorithm discussed before. When transitioning to multi-sentence tasks, small modifications are necessary. Specifically, for complex tasks such as question answering, the context sentence can be excessively long, preventing our prompt from fitting LLM's input limit. Even when the prompt fits, generating rationales for each context sentence can be prohibitively costly. Hence, for these situations, we resort to a more traditional seed data synthesis approach.

Specifically, we perform dataset synthesis given a set of conditional contexts $\mathcal{C} = \boldsymbol{c}_1, \cdots, \boldsymbol{c}_m$ (e.g., premise in NLI and context & answer in QA task). We perform dataset synthesis as follows:

1. Uniformly sample the current context $\boldsymbol{c}_{curr}$ sentence from $\mathcal{C}$, and current target label $y_{curr}$ from all possible labels $\mathcal{Y}$. Combine them into a seed data synthesis prompt $\mathcal{T}_{query}^{(2)}(\boldsymbol{c}_{curr}, y_{curr})$.

2. Synthesize the target sentence (e.g., hypothesis in NLI and question in QA) from LLM by $\mathcal{T}_{query}^{(2)}(\boldsymbol{c}_{curr}, y_{curr})$. The synthesized data is denoted as $(\boldsymbol{c}_{curr}, x_{syn}, y_{curr})$.

3. Repeat the above steps until we have enough seed data $\mathcal{D}_{seed} = (\mathcal{C}_{seed}, \mathcal{X}_{seed}, \mathcal{Y}_{seed})$

| Dataset | Prompt Type | Prompt | Label word (Y) |
|---|---|---|---|
| IMDb | $\mathcal{T}_{ration}$ | Imagine you are watching a movie; consider <X> reasons that may lead to <Y> impression of the movie. | positive/ negative |
| | $\mathcal{T}_{query}^{(1)}$ | Now imagine that you just watched a movie that has <X>. Now you should write a <Y> review about this movie. | positive/ negative |
| | $\mathcal{T}_{mis}^{(1)}$ | Write a <Y> movie similar to: \n <X> | positive/ negative |
| QNLI | $\mathcal{T}_{query}^{(2)}$ | Given an information paragraph: <X> \n Please ask a question that has answers <Y> the information paragraph | in/ not in |
| | $\mathcal{T}_{mis}^{(2)}$ | Given a premise: <X["premise"]> \n And here is a question: <X["question"]> that the answer of question is <Y> the premise.\nPlease write another question similar to the given question and have answers <Y> the premise. | in/ not in |
| RTE | $\mathcal{T}_{query}^{(2)}$ | <X> \nBased on the above description, the following sentence is definitely <Y>: | correct/ wrong |
| | $\mathcal{T}_{mis}^{(2)}$ | <X["premise"]> \nBased on the above description, the following sentence: <X["Hypothesis"]> is definitely <Y>. Now write a sentence similar to the given sentence and is definitely <Y> based on the given description. | correct/ wrong |
| AdQA | $\mathcal{T}_{query}^{(2)}$ | Given a context: <X["context"]> \nX<["answer"] is the answer to the following question: | NA |
| | $\mathcal{T}_{mis}^{(2)}$ | Given a context: <X["context"]> \nX<["answer"] is the answer to: <X["question"]>.\nA question that has the same answer in the context is: | NA |

Table 1: Designed prompts for the four datasets. $\mathcal{T}_{ration}$ denotes the prompt for the LLM to generate rationales. $\mathcal{T}_{query}^{(1/2)}$ denotes the prompt for seed data synthesis, and <X> denotes the rationale list or context sentences for the current seed data example. $\mathcal{T}_{mis}^{(1/2)}$ denotes the prompt for EES, where <X> is the full misclassified example.

For the EES process, in multi-sentence tasks, we only need to modify the for-loop beginning at line 8 in Alg. 2 to fit the multi-sentence task. The changed version of line 8 is shown in Alg. 3.

---

**Algorithm 3:** Multi-sentence EES, inner for-loop

---

1 **for** *each* $(c_{mis}, x_{mis}, y_{mis}) \in \mathcal{D}_{mis}^{(q)}$ **do**
2    $x_{add} \sim \mathbb{P}_{LLM}(\cdot | \mathcal{T}_{mis}^{(2)}(c_{mis}, x_{mis}, y_{mis}))$
3    $\mathcal{D}_{add}^{(q+1)} \leftarrow \mathcal{D}_{add}^{(q+1)} \cup \{(c_{mis}, x_{add}, y_{mis})\}$

---

## 2.5 Prompt engineering

The design of prompts can have a huge impact on the quality of the synthesized dataset. We present the prompt templates used for generating rationales, data points, and error extrapolation in Table 1.

## 2.6 Theoretical Analysis

In this section, we give a detailed analysis of why our S3 framework can shrink the distribution gap between zero-shot synthesis and real-world distribution by first clarifying the analysis setup and then giving an analysis of the distribution gap problem and the effectiveness of our S3 framework.

We denote the probability space of the data example as $\mathcal{P} = (\mathcal{S}, \Sigma)$; here, for simplicity, we wrap all possible elements in a data example into one variable $s \in \mathcal{S}$, and the components in $s$ can be varied depending on the specific task, for example, in the text classification task, i.e., $s = (x, y)$ where $x$ is a piece of text and $y$ is the corresponding label.

We assume that the gold dataset (denoted as $\{S_i^{(gold)}\}_{i=1}^{n_{gold}}$) is obtained by i.i.d. sampling $n_{gold}$ times from a real-world distribution $\mathbb{P}_{\mathcal{D}} \in \mathcal{P}$. Then, we also assume the process of obtaining a synthesized data example as an i.i.d sampling from $\mathbb{P}_{LLM} \in \mathcal{P}$. In the analysis section, for simplicity, we define $\mathbb{P}_{LLM}$ as a distribution over the data example set $\mathcal{S}$ instead of the space of human language. This distinction is important because while text data is in natural language, for many tasks, labels may not be.

Similarly, we assume that the process of attaining the seed dataset (denoted as $\{S_i\}_{i=1}^{n_1}$), where $n_1$ is the number of seed data points, is to draw $n_1$ i.i.d. samples from our seed data distribution

$\mathbb{P}_{LLM}^{(0)}$.

Let us first recall the origin of the distribution gap problem in dataset synthesis methods: conventional data synthesis methods, as well as the seed dataset synthesis stage in our approach, sample data points from a fixed distribution $\mathbb{P}_{LLM}^{(0)}$. Since the distribution is fixed and different from the task data distribution $\mathbb{P}_{\mathcal{D}}$, the synthesized dataset suffers from a fixed distribution gap no matter how much data we synthesize. Therefore, the testing performance of the small model trained on the synthesized dataset on real task data is bounded by this gap. Our approach, S3, aims to resolve this limitation.

Let us assume that the small model perfectly learns the synthesized dataset distribution. In this case, the error that the small model makes on the small gold validation dataset can represent the distribution gap between $\mathbb{P}_{\mathcal{D}}$ and $\mathbb{P}_{LLM}^{(0)}$.

Finally, we argue that a good LLM can perfectly extrapolate from the errors. This means that the LLM can synthesize samples from the difference between two distributions $\mathbb{P}_{\mathcal{D}} - \mathbb{P}_{LLM}^{(0)}$. Formally, the additional data synthesized in each round of the EES process follows:

$$\mathbb{P}_{add} := \mathbb{P}_{LLM}(\cdot | \mathbb{P}_{\mathcal{D}} - \mathbb{P}_{LLM}^{(0)}) \qquad (1)$$

Therefore, by sampling the same number of data points from $P_{add}$ and combining them with the original seed data distribution $P_{LLM}^{(0)}$, the mixed dataset shall follow the distribution:

$$\mathbb{P}_{LLM}^{(1)} := p \cdot \mathbb{P}_{add} + (1 - p)\mathbb{P}_{LLM}^{(0)} \approx \mathbb{P}_{\mathcal{D}} \quad (2)$$

where $p \in [0, 1]$ is the ratio of combination, it can be intuitively understood as the portion of the additional dataset and seed dataset. This suggests that, theoretically, we can recover the gold data distribution by simply combining the original seed data and the additional data synthesized via EES.

However, please note that we cannot guarantee the LLM and the training of the small model are perfect in real-world scenarios. Therefore, S3 repeats this process iteratively to gradually reduce the distribution gap and optimize the mixed dataset until convergence.

## 3  Experiments

We conduct experiments to test the effectiveness of our approach across three major NLP tasks over four datasets. We also do a thorough ablation study (Section 3.4), a transferability study (Section 3.5) for the S3 framework, and a study on additional data quality (Section 3.6).

### 3.1  Setup

#### 3.1.1  Datasets

In this study, we evaluate our S3 on three major NLP tasks: text classification, Natural Language Inference (NLI), and Question Answering (QA). For text classification, we use the IMDb (Maas et al., 2011) dataset; for the NLI task, we use the QNLI (Rajpurkar et al., 2016; Wang et al., 2018) and the RTE (Bentivogli et al., 2009; Giampiccolo et al., 2007; Haim et al., 2006) dataset; for the QA task, we use the Adversarial QA (Bartolo et al., 2020) dataset.

### 3.2  Baselines

We compare our S3 framework with the following baselines:

1. **ZeroGen:** ZeroGen is the basic data synthesis method proposed by Ye et al. (2022b). It neither uses rationales for data synthesis nor attempts to reduce the distribution gap. Note that ZeroGen also uses the same small validation set for tuning hyperparameters.

2. **GoldGen:** This baseline extrapolates the entire gold validation data instead of the errors made by the small model. We further use this baseline to test the effectiveness of the error extrapolation idea in the S3 framework. We keep the scale of synthesized datasets the same in order to make a fair comparison with S3.

3. **ProGen:** This baseline was proposed by Ye et al. (2022a), like the EES, it also considers training feedback. However, this framework is only available for text classification tasks, and it does not use LLM rationales for data synthesis.

4. **Gold Data:** We also include a baseline that trains the small model on the original gold data for reference.

#### 3.2.1  Implementation details

This section gives full implementation details of S3 in our experiments. We apply GPT3.5 derived from (Brown et al., 2020) as the LLM for all the synthesis work, and we use nucleus sampling (Holtzman

| Method | Data Size / Results | IMDb | QNLI | RTE | Adversarial QA (EM/F1) | Average |
|---|---|---|---|---|---|---|
| Gold Data | Data Size | *25k* | *105k* | *2.5k* | *30k* | *40.63k* |
| | Results | 87.93 | 88.05 | 58.12 | 18.6/29.85 | 56.51 |
| ProGen | Data Size | *100k* | - | - | - | - |
| | Results | 84.12 | - | - | - | - |
| ZeroGen | Data Size | *200k* | *200k* | *200k* | *200k* | *200k* |
| | Results | 84.28 | 71.19 | 59.93 | 6.33/9.96 | 46.34 |
| GoldGen | Data Size | *25k* | *150k* | *30k* | *80k* | *61.25k* |
| | Results | 87.93 | 78.31 | 64.25 | 11.63/23.33 | 53.09 |
| S3 | Data Size | *21.2k* | *168k* | *33.6k* | *81.5k* | *76.08k* |
| | Results | **89.00** | **79.92** | **73.29** | **12.50/24.38** | **55.73** |

Table 2: Main experimental results. All compared methods are evaluated by fine-tuning DistilBERT. The performance of fine-tuning the small model on gold data is in gray because it is not directly comparable with other results.

et al., 2019) with a temperature of 0.9 for decoding. We use DistilBERT-base-uncased (Sanh et al., 2020) provided by the Hugging Face Transformers library (Wolf et al., 2019) as the small model. We perform hyperparameter tuning on the batch size, learning rate, weight decay, and the number of epochs for fine-tuning the small model.

### 3.2.2 Evaluation Method

For text classification and NLI tasks, we use the *accuracy rate* as the evaluation method. For QA tasks, we use *Exact Match* (EM) and *F1 score* as evaluation methods. To implement the experiment of S3 method, we utilize the training data from the original dataset as the gold evaluation data dataset in EES (i.e., $\mathcal{D}_{gold}^{(eval)}$). And we use testing data from the original dataset to test our model's performance.

### 3.3 Experimental Results

We present our main experimental results in Table 2. We can observe that our S3 framework has a huge improvement (an average improvement of 9.48%) compared to ZeroGen. The performance gap is especially large in NLI and QA tasks. Moreover, we only use an average of 30.43% amount of data compared to ZeroGen, which can be considered as a significant improvement. Such an improvement proves the effectiveness of the initial seed data synthesis method and the idea to keep on optimizing the data in our S3.

We then compare S3 with the GoldGen baseline to test the effectiveness of extrapolating the errors of the small model on the validation set in-

stead of the entire validation set. We find that S3 outperforms GoldGen with an average absolute performance improvement of 2.73%. This confirms the advantage of error extrapolation over directly extrapolating gold data.

It is also noteworthy that S3 yields competitive results compared to directly fine-tuning the small model on the full gold training data. Specifically, S3 even outperforms gold data performance on IMDB and RTE. This confirms the potential of applying S3 in real-world applications.

### 3.4 Ablation Study

#### 3.4.1 Ablation of EES

We first ablate the error extrapolation-based synthesis (EES) framework of S3, using only the seed data synthesized based on Section 2.2. We make sure that the scale of the training dataset is approximately the same for a fair comparison. The result can be seen in Table 3. This result proves the effectiveness of our view of the dynamic dataset and EES. We find that for more complex tasks like QA and NLI, our EES framework can give a larger improvement, which proves the distribution gap problem and our EES framework's ability to shrink this gap.

#### 3.4.2 Ablation of Seed Data Synthesis with Rationales

We then ablate the use of rationale for dataset synthesis in the S3 framework on the IMDb dataset. The results are shown in Table 4. We find that using rationale for dataset synthesis enables the LLM to generate datasets of higher quality that leads to

| Method | IMDb | QNLI | RTE | Adversarial QA |
|--------|------|------|-----|----------------|
| S3 | **89.00** | **79.92** | **73.29** | **12.50/24.38** |
| w/o EES | 86.86 | 73.70 | 65.71 | 8.70/20.03 |

Table 3: Ablation test results (%) on iterative error extrapolation. The baseline w/o error extrapolation is fine-tuned on the same amount of data compared to S3.

better performance of the small model with a lower budget, i.e., fewer synthesized examples.

| | with Rationale | w/o Rationale |
|--------|----------------|---------------|
| Dataset Size | *15k* | *40k* |
| Results (%) | **86.86** | 85.34 |

Table 4: Experiment result of ablation of rationales analysis in seed data synthesis. The section with Rationale means we synthesize seed data guided by a set of LLM synthesized rationales, and w/o Rationale means the seed data is synthesized by the task-descriptive prompt without rationale.

### 3.5 Transferability of EES Data

We then test the transferability of the EES-synthesized data. The results are shown in Table 5. In this test, we replace the seed dataset of our framework with the data synthesized by Ye et al. (2022b). We do two sets of testing. We compare the variants where we directly add the EES data synthesized in S3 (+ourAdd) and that with the small model trained on the data synthesized by Ye et al. (2022b). We can see that the two variants both lead to similar performance improvements. This shows that the EES synthesized data can effectively transfer to other zero-shot synthesized datasets. We believe this is because the distributional gap for different zero-shot data synthesis methods is similar. Therefore, the data synthesized by the EES method can be universally helpful, which further demonstrates the potential of S3.

| Method | IMDb | QNLI | AdQA |
|--------|------|------|------|
| ZeroGen | 84.28 | 68.60 | 4.60/9.62 |
| +ourAdd | 87.50 | 73.51 | 9.70/20.10 |
| +synAdd | 87.41 | 72.21 | 10.27/19.92 |

Table 5: Transferability test result (%): where +ourAdd is ZeroGen dataset as seed data and S3 synthesized data as additional data, and +synAdd is using EES on ZeroGen trained small model's misclassified data

### 3.6 Additional data quality study

We perform this experiment to check the quality of the additional dataset synthesized by EES. Note that for earlier LLMs like GPT2 (Radford et al., 2019) or T5 (Raffel et al., 2020), there used to be a tendency to repeat the prompt. If the LLM just repeats the misclassified data, then there is no extrapolation. Thus, we composed experiments as follows to test the quality of the additional dataset:

**Sentence Encoding:** For both misclassified data $\mathcal{D}_{mis}$ and additional data $\mathcal{D}_{add}$, we use DistilBERT to encode each $x_{mis}$ and $x_{add}$. This results in encoded sentences represented as $z_{mis}$ and $z_{add}$ respectively, and each encoded sentence is in $\mathbb{R}^d$ (with $d = 768$ in DistilBERT)

**Cosine Similarity:** Then, by comparing the cosine similarity between $z_{mis}$ and $z_{add}$, we gauge their semantic similarity. High cosine similarity indicates substantial semantic overlap.

**Edit Distance:** Further, to understand textual distinctiveness, we compute the edit distance between sentences $x_{mis}$ and $x_{add}$. If the edit distance approaches the sentence length, we infer that the texts differ significantly in their composition. The results are shown in Table 6.

| Label | IMDb | QNLI | RTE | AdQA |
|-------|------|------|-----|------|
| Data Num | 6,173 | 51,100 | 1,522 | 51,532 |
| Avg. Cos Sim | 0.9497 | 0.9537 | 0.9380 | 0.9468 |
| Avg. Edit Dist. | 273.92 | 14.64 | 16.38 | 13.99 |
| Avg. $x_{mis}$ len | 288.04 | 14.17 | 13.91 | 13.73 |
| AVG. $x_{add}$ len | 218.72 | 19.97 | 24.61 | 18.70 |

Table 6: Quality study of Additional Data

The average misclassified data length (avg $x_{mis}$ len) and average generated data length (avg $x_{add}$ len) provide context to interpret edit distances. This result shows that while there is high semantic similarity among the misclassified data and the additional generated data (evidenced by the cosine similarity scores), the generated sentences are not mere copies of the misclassified samples (as their edit distance is almost the length of the whole sentence). This result provides extra evidence in favor of the quality of the newly generated data.

## 4 Related work

### 4.1 Dataset Synthesis

The vast quantity of data required by the majority of Machine Learning methodologies has prompted numerous researchers to explore the concept of *Dataset Synthesis*. This aims to generate a dataset

from large pre-trained models, such as LLMs, in order to transfer rich knowledge from large models to small models. Initial attempts to achieve this used fine-tuned generative models to generate data (Anaby-Tavor et al., 2020; Kumar et al., 2020). These efforts involved first fine-tuning the LLMs with a small amount of human-annotated data (gold data), then combining the generated data with gold data to train small models. Other researchers sought to synthesize copious amounts of data for semi-supervised learning (Chen et al., 2020; Wang et al., 2021). Nonetheless, these methods are only suitable for straightforward text classification tasks, proving data inefficient and ineffective for more complex tasks like NLI or QA.

The potential of zero-shot performance offered by LLMs has led some researchers to consider zero-shot dataset synthesis based on non-finetuned LLMs (Meng et al., 2022; Ye et al., 2022b). However, as indicated by Fig1, direct querying of non-fine-tuned LLMs often results in data that suffers from a large distribution gap and is typically inefficient. Thus, some studies have attempted data selection (Gao et al., 2023) or data augmentation (Ye et al., 2022a). However, their capacity to rectify the distribution gap leaves room for improvement.

### 4.2 In-context Learning

Brown et al. (2020) suggests LLMs can better learn the task they are working on by conditioning on a few examples in the prompt. This paradigm, known as *In-context learning*, is particularly appealing as it negates the necessity of updating the parameters of LLM. Subsequent research has focused on optimizing the choice of prompt templates and in-context examples (Liu et al., 2021; Wang et al., 2023; Lu et al., 2021), and learning with in-context objective descriptions (Chen et al., 2021). The key idea for in-context learning is to learn from analogy (Dong et al., 2022), which aligns with our idea of extrapolating error to synthesize additional data to fill the distribution gap. However, most in-context learning methods are designed for a few-shot setting, whereas in our research, the LLM does not need to be trained. We explore the LLM's ability to directly extrapolate from errors, providing a crucial example for creating a more effective dataset.

## 5 Conclusion

This paper proposes the *Synthesis Step by Step* (**S3**) approach based on a *dynamic dataset* viewpoint

for dataset synthesis. **S3** is a novel dataset synthesis framework that shrinks the distribution gap between purely LLMs synthesized datasets and the real underlying data distribution. S3 achieves this by first using seed data synthesis with rationales to have a low distribution gap in seed data. It shrinks this distribution gap by iteratively extrapolating errors of the small model on a small amount of real-world data. Extensive experiments on three major NLP tasks over four commonly used datasets show that compared with a representative baseline, S3 significantly improves the performance of a small model with averagely only one-third of synthesized data. S3 has high practical potential in many real-world applications because it can effectively (i.e, with better performance) and efficiently (i.e., with improved data efficiency) transfer knowledge in an extremely large model (e.g., GPT 3.5) to a small model (e.g., DistilBert), achieving data efficiency and computation efficiency at the same time.

## Acknowledgments

We thank the anonymous reviewers for their feedback on our paper. MS acknowledges support from the Swiss National Science Foundation (Project No. 197155), a Responsible AI grant by the Haslerstiftung; and an ETH Grant (ETH-19 21-1).

## Limitations

Although S3 achieved promising results, there are still several limitations of our work. The first limitation is that in the experiments, we spotted that a tiny change in the synthesis prompts can lead to a significant performance drop, which means our framework is not prompt-stable. A possible future direction is to develop a systematic way to compose prompts that can perform stably well by fine-tuning an LLM using good prompts. The second limitation is that S3 assumes that the LLM has a rich knowledge of the specific task. But in the actual application of the approach in the real-world, there is no such guarantee. A possible solution to mitigate this limitation is to ask the LLM to divide the previously unseen task into multiple simple tasks that the LLM has a good understanding of, but it also requires the LLM to have a good ability to understand the subtasks. The third limitation is that S3 is task-specific. Future work may try to extend the method to cross-task settings to further improve the computational and data efficiency of the method.

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

## A   Intuitive understanding to EES

Since the pseudo-code of EES may be somewhat non-intuitive to understand, this part aims to provide an intuitive understanding of the EES method on single-sentence tasks.

## A.1 Attain Error

The first step for EES is to attain the error made by the small model on the gold validation dataset, which is, to a certain extent, the representation of the distribution gap between LLM's seed data synthesis distribution and the real-world distribution. To attain the error, we must first train the small model with currently synthesized data. This includes the seed data $\mathcal{D}_{seed}$, and additional datasets $\mathcal{D}_{add}^{(0)}, \cdots, \mathcal{D}_{add}^{(q)}$, where $q$ is the current round of iteration. Then we have $\mathcal{D}_{add}^{(0)} = \emptyset$. Thus, the training dataset for $q$-th iteration is:

$$\mathcal{D}_{train}^{(q)} = \mathcal{D}_{seed} \cup (\cup_{j=0}^{q} \mathcal{D}_{add}^{(j)}) \qquad (3)$$

Then, we train the small model with $\mathcal{D}_{train}^{(q)}$. We denote the fitted small model as $f(\cdot | \mathcal{D}_{train}^{(q)})$. Then, we evaluate the fitted small model on the gold validation dataset and obtain the data samples with high error in the validation dataset:

$$\mathcal{D}_{mis}^{(q)} = misclass\{f(\mathcal{D}_{gold}^{(eval)} | \mathcal{D}_{train})\} \qquad (4)$$

where the $misclass$ function denotes the function that attains the data samples that have been misclassified. For instance, for the QA task, this can mean data samples that do not have an exact match with the answer or data samples with low F1 scores. We represent the distribution gap between the underlying truth and the $\mathcal{D}_{train}^{(q)}$ by the misclassified gold evaluation dataset $\mathcal{D}_{mis}^{(q)}$, which is the distribution gap in $q$-th round of EES.

## A.2 Synthesis on extrapolating error

After having $\mathcal{D}_{mis}^{(q)}$, for all the misclassified data $(x_{mis}, y_{mis}) \in \mathcal{D}_{mis}^{(q)}$, we query the LLM again using a prompt that wraps information of the misclassified data. The prompt $\mathcal{T}_{mis}^{(1)}(x_{mis}, y_{mis})$ intuitively asks the LLM to extrapolate the misclassified data and synthesize a new data example. For example, in the movie classification problem, if the current misclassified data is: (*The move is great, positive*); our original $f(\cdot | \mathcal{D}_{train}^{(q)})$ labeled such a review as a negative one. In this case, $\mathcal{T}_{mis}^{(1)}(x_{mis}, y_{mis})$ can be something like *Generate a **positive** movie review like **The move is great***.

We query the LLM with $\mathcal{T}_{mis}^{(1)}(x_{mis}, y_{mis})$, to obtain another data example similar to the error. This process is repeated for every misclassified data point. Thus, we obtain the $q + 1$-th additional dataset $\mathcal{D}_{add}^{(q+1)}$. We repeat the *Attain Error* and *Synthesis on extrapolating error* steps for multiple rounds until the error converges. With such a method, we can optimize our synthesized dataset step by step to attain a dataset with a lower distribution gap by utilizing the information provided by extrapolating errors that represent the distribution gap.

## B Computation complexity comparison between S3 and ZeroGen

This section studies the total computation cost of the S3 framework. We compare the number of floating-point operations (FLOPs) involved in fine-tuning the model with S3 and ZeroGen synthesized dataset. For the BERT family of models, according to Brown et al. (2020), they cost 6 FLOPs per token per parameter (i.e., $F_{token,para} = 6$) in training. The DistilBERT model (Sanh et al., 2020) has $n_{para} = 66 \times 10^6$ parameters and the typical input length for one record is $num_{rec}^{(token)} = 512$. Therefore, the training FLOPs per record of data per epoch is:

$$
\begin{aligned}
F_{rec} &= F_{token,para} * num_{rec}^{(token)} * n_{para} \\
&= 2.03 \times 10^{11}
\end{aligned}
$$

The ZeroGen method typically uses $200k$ records of data and trains for an average of 10 epochs to achieve the best results based on our experiments. Thus, the total fine-tuning cost in terms of FLOPs for ZeroGen is:

$$F_{ZeroGen} = F_{rec} * 200k * 10 = 4.06 * 10^{17}$$

In S3, in the first round of fine-tuning (using only the seed data), the dataset size is $51.2k$ records on average (i.e., seed dataset is about $2/3^{th}$ size of final dataset). After one round of EES, the total dataset size becomes $64.0k$ (i.e., $5/6^{th}$ size of the final dataset). The final round of fine-tuning with two EES additional datasets and the seed dataset that have a total size of $76.8k$ records of data. On average, our method needs 8 epochs to achieve its best result. Therefore, the total number of FLOPs of fine-tuning DistilBERT for the 3 iterations (2 for getting misclassified data, 1 for final fine-tuning) in our S3 is:

$$
\begin{aligned}
F_{S3} &= F_{rec} * (51.2k + 64.0k + 76.8k) * 8 \\
&= 3.11 * 10^{17}
\end{aligned}
$$

| label | Generated Movie Review |
|---|---|
| Negative | If you're looking for a film that will make you cringe in disbelief, look no further than "Zombie Cheerleader Apocalypse." This movie is the epitome of a low-budget, poorly-made production. The acting is atrocious, and the jokes are so bad that they will make you want to gouge your eyes out. The story revolves around a group of cheerleaders who are turned into zombies after a failed science experiment. They then proceed to wreak havoc on their school and town, killing anyone in their path. The supposed "hero" of the film is a nerdy guy who has a crush on the head cheerleader and tries to save her from the zombie apocalypse. The special effects are laughable, and the makeup for the zombies is so poorly done that it's hard to tell if they're supposed to be scary or comical. The dialogue is cringeworthy, and the attempts at humor fall completely flat. The film tries to be a parody of horror movies, but instead comes off as a cheap knockoff. Overall, "Zombie Cheerleader Apocalypse" is a complete waste of time. It's not even bad in a fun way, it's just plain bad. The only reason to watch this movie is if you want to torture yourself for 90 minutes, otherwise, avoid it at all costs. |
| Positive | "The Family Man" is a thrilling and heartwarming movie that will keep you at the edge of your seat. The movie revolves around Srikant Tiwari (Manoj Bajpayee), a senior analyst in the National Intelligence Agency, who is trying to balance his demanding job and his personal life with his wife Suchitra (Priyamani), and their two children. The movie showcases the struggles of maintaining normalcy in Srikant's life, while also juggling intense missions. The action sequences are brilliantly choreographed, with twists and turns that keep you guessing till the very end. Bajpayee's performance is outstanding, as he brings a depth and complexity to his character. The chemistry between him and Priyamani is natural and heartwarming, adding a layer of emotion to the story. The supporting cast is equally impressive, with Sharib Hashmi and Shreya Dhanwanthary delivering memorable performances. The movie also explores important themes such as the impact of terrorism, family values, and duty towards one's country. The director, Raj Nidimoru and Krishna D.K., have done a remarkable job of balancing the action with the human story, making it a compelling watch for all audiences. Overall, "The Family Man" is a must-watch for anyone looking for a thrilling and entertaining ride, with a strong message of family and patriotism." |
| Positive | If you're a rom-com fan, then you'll love this movie! It's a fun, light-hearted take on the classic boy-meets-girl trope, with plenty of laughs and heartwarming moments to keep you entertained. The chemistry between the two leads is great, and the supporting cast is equally charming. While some may criticize the predictable plot and lack of originality, I found it to be a comforting and enjoyable watch. The soundtrack is also a standout, with catchy songs that perfectly capture the mood of each scene. I highly recommend this movie for anyone looking for a feel-good, easy watch. 8/10. |
| Negative | This movie is a complete waste of time. The story is nonexistent and feels like the director just slapped together random scenes. The characters are forgettable and lack any depth or development. The pacing is so fast that it's hard to keep up with what's going on. It's like watching a poorly made music video on repeat.The only redeeming quality of this film is the cinematography, but even that can't save it from being a complete mess. It's clear that the director has some talent behind the camera, but they need to work on their storytelling skills.Overall, I wouldn't recommend this film to anyone. Save your money and go watch something else. Let's hope the director can learn from their mistakes and create something better in the future. |

Table 7: Examples of generated IMDb data

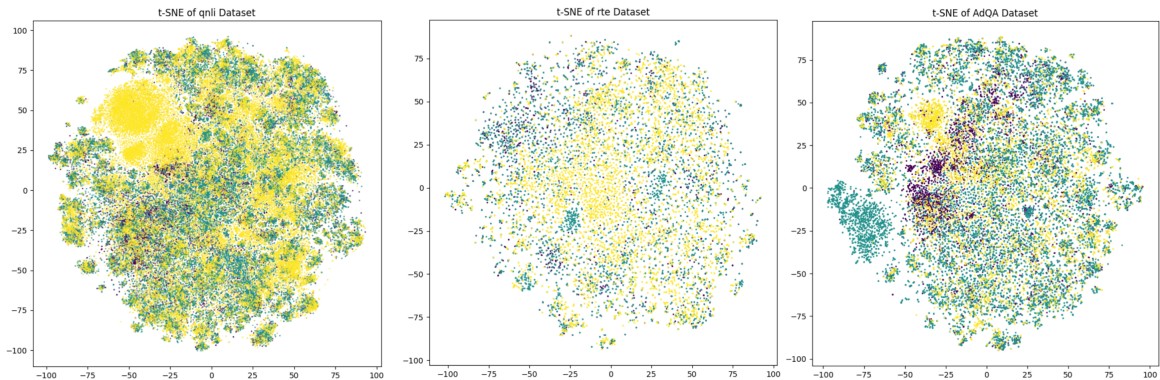

Figure 3: t-SNE result for QNLI (left), RTE (center), AdQA (right) for dataset diversity analysis. ZeroGen data's points are plotted in Yellow, S3's in Green, and Gold data in Purple.

| Dataset | S3 Coverage | ZeroGen Coverage |
|---------|-------------|------------------|
| QNLI | 76.35 | 63.03 |
| RTE | 73.59 | 14.90 |
| AdQA | 51.02 | 46.00 |

Table 8: Coverage rate (%) of S3 and ZeroGen

| Method | IMDb | QNLI | RTE | AdQA |
|--------|------|------|-----|------|
| Gold Data | 92.30 | 91.00 | 71.50 | 22.97/36.59 |
| ZeroGen | 83.66 | 70.11 | 72.2 | 5.07/10.74 |
| S3 | 89.55 | 85.20 | 76.17 | 20.50/34.40 |

Table 9: Apply S3 framework on MiniLM

To conclude, due to fewer rounds of fine-tuning epochs and the lower need for data, S3 uses only $3/4^{th}$ the number of FLOPs compared to the Zero-Gen baseline, even though we fine-tuned the model multiple times.

## C   Dataset Diversity analysis for S3

This section analyzes the diversity of the synthesized sentences. Such an analysis is necessary as the LLMs may generate sentences with similar meanings, rendering the dataset lacking in diversity. As there is no universally approved method for analyzing dataset diversity, we use both quantitative and qualitative methods to analyze dataset diversity:

### C.1   Quantitative Analysis:

For short synthesized sentences, such as the QNLI, RTE, and AdQA datasets, we approach the dataset analysis quantitatively. Given the high hidden dimension of the sentence encoding (e.g., 768 for DistilBERT), direct analysis can be inefficient. Hence, we used t-SNE for dimension reduction (Van der Maaten and Hinton, 2008). The final steps of our analysis are as follows:

1. Uniformly sample a similar amount of data from gold data, S3 synthesized data, Zero-Gen synthesized data. We have $\mathcal{D}'_{gold} = \{x^{(i)}_{gold}, y^{(i)}_{gold}\}^{n_1}_{i=1}$, $\mathcal{D}'_{S3} = \{x^{(j)}_{S3}, y^{(j)}_{S3}\}^{n_2}_{j=1}$, and $\mathcal{D}'_{ZeroGen} = \{x^{(k)}_{ZeroGen}, y^{(k)}_{ZeroGen}\}^{n_3}_{k=1}$ where $n_1, n_2, n_3$ should be similar.

2. Encode the sentences using DistilBERT. Then, we have the sentence encodings: $\{z^{(i)}_{gold}\}^{n_1}_{i=1}, \{z^{(j)}_{S3}\}^{n_2}_{j=1}, \{z^{(k)}_{ZeroGen}\}^{n_3}_{k=1} \subseteq \mathbb{R}^d$, where $d$ is the hidden state's dimension (in our case, it is 768).

3. Perform t-SNE on the encoded data $\boldsymbol{z} := \{z^{(i)}_{gold}\}^{n_1}_{i=1} \cup \{z^{(j)}_{S3}\}^{n_2}_{j=1} \cup \{z^{(k)}_{ZeroGen}\}^{n_3}_{k=1}$ to reduce the dimension from $d$ to 2. We have: $t-SNE(\boldsymbol{z}) = \boldsymbol{p} = \{p^{(i)}_{gold}\}^{n_1}_{i=1} \cup \{p^{(j)}_{S3}\}^{n_2}_{j=1} \cup \{p^{(k)}_{ZeroGen}\}^{n_3}_{k=1} \subseteq \mathbb{R}^2$

4. Draw the reduced dimension points on a scatter plot to directly see the overlap of our synthesized dataset and the Gold data. We show the results in Fig. 3. We can see that the green region significantly aligns with the purple region, which indicates that S3 results in similar data diversity as the gold data.

Data diversity can also be quantified by counting how many points of $p^{(k)}_{gold}$ are in the area of $A_{S3} := \cup^{n_2}_{j=1} B_\gamma(p^{(j)}_{S3})$ and $A_{ZeroGen} := \cup^{n_3}_{k=1} B_\gamma(p^{(k)}_{ZeroGen})$, where $B_\gamma(p)$ represents a solid circle with center $p$ and radius $\gamma$. The results for QNLI, RTE, and AdQA are shown in Table 8.

The results further demonstrate the superior coverage and diversity of our S3 framework compared to ZeroGen.

## C.2 Qualitative Analysis:

For tasks that require the generation of longer texts, the text encoding approach is not amenable to t-SNE dimension reduction and interpretation. Thus, in such settings, we conduct qualitative analysis. We show examples of the generated data for the case of sentiment classification of IMDB reviews in Table 7. We can observe that these examples exhibit rich contexts and diverse patterns, which supports the superiority of our S3 framework. For more qualitative results, please refer to the dataset in our project repository.

## D Additional Results for S3 with MiniLM

In addition to DistilBERT, we also evaluated the performance of the Synthesis Step by Step (S3) framework using MiniLM (Wang et al., 2020) as the small model. The results of this experiment are presented in Table 9. Notably, there is a substantial enhancement in performance when compared to the ZeroGen baseline in all the tasks. Moreover, in tasks like RTE which lack data, our method even surpasses the performance of the model trained on gold data. These results provide robust evidence that the effectiveness of S3 is not limited to a specific model. Instead, it offers consistent improvements across different small models, underscoring its broad applicability and efficacy.