# OpenReview forum: "Let's Synthesize Step by Step: Iterative Dataset Synthesis with Large Language Models by Extrapolating Errors from Small Models"
_EMNLP/2023/Conference — EMNLP 2023 Findings_

### Official Review · Reviewer_fvWF · 2023-08-04

**Soundness:** 4

**Excitement:**

3: Ambivalent: It has merits (e.g., it reports state-of-the-art results, the idea is nice), but there are key weaknesses (e.g., it describes incremental work), and it can significantly benefit from another round of revision. However, I won't object to accepting it if my co-reviewers champion it.

**Paper Topic And Main Contributions:**

This work proposes a data synthesize framework S3 that uses a large language model to generate prompt templates with rational information and construct a seed dataset, which is then used to train a smaller model. The seed dataset is iteratively augmented by generating additional data that are misclassified from a small real dataset. Experimental results show that S3 achieves better improvement over previous data synthesize frameworks like ProGen and ZeroGen. And the template with rational information achieves similar performance compared to augmentation without the rational template and using only 30% of the data. Moreover, S3 has better transferrability to other synthetic dataset.

**Reasons To Accept:**

- A simple and effective data synthesize framework that mitigates the demand of data generation (associated with number of queries to large language models) using rational prompt templates.
- Some qualitative analysis to demonstrate why the iterative process of S3 can mitigate the distribution gap between synthesized data and real data.

**Reasons To Reject:**

- The incorporation of rationals can be challenging for tasks beyond classification of standard QA tasks (fluency, consistency, factuality etc.), the work does not address the quality or includes a gating step to check the quality of synthesized data generated from misclassified examples.
- Since the small model needs to be finetuned iteratively from the synthesized dataset, it can be expensive by a large number of iterations, the information about the finetuning choices in this regard is not well discussed.

**Reproducibility:**

N/A: Doesn't apply, since the paper does not include empirical results.

**Reviewer Confidence:**

3: Pretty sure, but there's a chance I missed something. Although I have a good feel for this area in general, I did not carefully check the paper's details, e.g., the math, experimental design, or novelty.

**Typos Grammar Style And Presentation Improvements:**

- Redundant parenthesis on the last line of Algorithm 1.
- The superscript of template in the caption of Table 1 is confusing.

---

> ### Author Rebuttal · Authors · 2023-08-29
>
> Dear Reviewer fvWF
>
> #### Answers to Reasons To Reject:
>
> ##### Response to the SSE synthesized data quality check:
>
> We agree that more sophisticated prompt design is required to apply rationale-based data synthesis on datasets beyond classification and QA. We will acknowledge this in the discussion and limitation section and will leave the application of our approach on more general tasks for future work.
>
> As for the checking of quality of synthesized data, we perform post-hoc checking to ensure the quality of the EES synthesized data. Here are the detailed steps:
>
> 1. For misclassified data $D_{mis} = \{(x_{mis}^{(i)}, y_{mis}^{(i)})\}$, $i$ is from 1 to n and additional data generated on the misclassied data $D_{add} = {(x_{mis}^{(i)}, y_{mis}^{(i)})\}$, $i$ is from 1 to n, we perform sentence encoding by DistilBERT on every $x$'s. We have encoded sentence $z_{mis}^{(i)} = enc(x_{mis}^{(i)}), z_{add}^{(i)} = enc(x_{add}^{(i)}) \in \mathbb{R}^{d}$ where $d$ is the hidden dimension, for DistilBERT $d = 768$.
> 2. After we have the encoded sentence for $ z_{mis}^{(i)}, z_{add}^{(i)}$ , we compare their cos similarity. If they have high cos similarity, it means they are similar in meaning.
> 3. Then, we compare the editing distance between sentence $x_{mis}^{(i)}, x_{add}^{(i)}$; if their editing distance is close to their sentence length, we say that they are significantly different in composing of text (i.e., they are not similar in text)
>
> By conducting the upon steps on the two rounds of generated on misclassification data, the result are as follows:
>
> | Rounds | Label | IMDb | QNLI | RTE | AdQA|
> |--------|-------|------|------|-----|-----|
> |Round 1 |data num | 3,291 | 28,532 | 883 | 26,605|
> |        |cos sim | 0.9445 | 0.9536 | 0.9367 | 0.9469 |
> |        |editing distance| 289.07 | 14.60 | 16.44 | 13.98 |
> |        |avg mis len| 282.45 | 14.20 | 13.75 | 13.76 |
> |        |avg gen len| 258.42 | 19.88 | 24.54 | 18.71 |
> |Round 2 |data num | 2,882 | 22,568 | 639 | 24,927 |
> |        |cos sim | 0.9557 | 0.9538 | 0.9397 | 0.9467|
> |        |editing distance | 265.61 | 14.68 | 16.29 | 14.01 |
> |        |avg mis len | 294.42 | 14.14 | 14.13 | 13.69 |
> |        |avg gen len | 173.40 | 20.09 | 24.70 | 18.69|
>
> We add the average length of misclassified data (avg mis len) and average length of generated data (avg gen len) for reference of the editing distance. In the chart, we can see that for all our extrapolation-based synthesis data, the cos similarity is high, which means they are close in meaning of the misclassified data. And their deiting distance is close to the length of data, which means the LLM does not simply copy from the misclassified data. That shows the new data generated is in high quality.
>
> ##### Responses to our framework’s fine-tuning cost:
>
> Thanks a lot for pointing out the potential high computation cost for our S3 framework caused by fine-tuning the small model iteratively. Let me address this by showing the fine-tuning cost of our framework is less than the ZeroGen baseline in sense of floating-point operations (FLOPs) involved in fine-tuning.
>
> For the DistilBERT model we used, based on OpenAI's report [1], the BERT family model costs $6$ FLOPs per training per token per parameter (i.e., $FLOP_{token, para} = 6$.). And based on [2], the DistilBERT has $numPara = 66 \times 10^6$ parameters. And the typical token length for one record is $ numToken_{rec} = 512$. Therefore, the training FLOP per record of data per epoch is:
>
> $$
> FLOP_{rec} = FLOP_{token, para} * numToken_{rec} * numPara = 2.03 * 10^{11}
> $$
>
> For ZeroGen, it typically uses $200k$ if data records and trains for an average $10$ epochs to achieve the best results based on our experiment. Thus, the fine-tuning cost for ZeroGen is:
>
> $$
> FLOP_{ZeroGen} = FLOP_{rec} * 200k * 10 = 4.06 * 10^{17}
> $$
>
> For our S3 method, in the first round of fine-tuning (using only seed data) dataset size is typically in $2/3$ size of final dataset (i.e. $51.2k$ records), and the second round of fine-tuning (using seed data + misclass generated data of first round) is in $5/6$ size of final dataset (i.e., $64.0k$), and the final round of fine-tuning (using seed data + 2 rounds of misclass generated data) have $76.8k$ data. And on average, our method needs $8$ epochs to achieve its best result. Therefore, the total FLOPs of fine-tuning DistilBERT for 3 times (2 for getting misclassified data, 1 for final fine-tuning) in our S3 frameworks are:
>
> $$
> FLOP_{S3} = FLOP_{rec} * (51.2k + 64.0k + 76.8k) * 8 = 3.11 * 10^{17}
> $$
>
> To conclude, we can see that our S3 methods use only 3/4 FLOPs compared to the ZeroGen baseline, even though we fine-tuned the model with R(=2) more times.
>
> #### Answers to other comments:
>
> We thank the reviewer a lot by detaily pointing out the problem we have in Alogrithm 1 and the subscript of template in the caption of Table 1. We have fixed them in the revised version.

---

### Official Review · Reviewer_s5eU · 2023-08-04

**Typos Grammar Style And Presentation Improvements:** 1. "Extrapolating the errors of the s…
**Soundness:** 4

**Excitement:**

3: Ambivalent: It has merits (e.g., it reports state-of-the-art results, the idea is nice), but there are key weaknesses (e.g., it describes incremental work), and it can significantly benefit from another round of revision. However, I won't object to accepting it if my co-reviewers champion it.

**Missing References:**

[1] Deep Active Learning for Biased Datasets via Fisher Kernel Self-Supervision, Denis Gudovskiy, Alec Hodgkinson, Takuya Yamaguchi, Sotaro Tsukizawa; Proceedings of the IEEE/CVF Conference on Computer Vision and Pattern Recognition (CVPR), 2020, pp. 9041-9049

**Paper Topic And Main Contributions:**

This paper proposes a method for progressively-improved multi-round synthetic data augmentation. The authors prompt LLMs to generate data samples given only labels/classes (and prompts), use synthetic data to train downstream model. Thereafter in each round, they generate more data specifically for labels which the downstream model misclassified. The authors method achieves large improvements across a range of datasets, using fewer synthetic samples than prior methods.

**Questions For The Authors:**

1. Did you test with R>2 rounds of training? (line 221)
1. Why is there such a difference in dataset size for GoldGen and S3 for QNLI (Table 1), when the authors say the size is kept comparable (line 377)? How is the performance of these models at equal size of data? (given that scores are very close)?
1. (Line 396) Could you specify the range of hyper-parameter search? Was the same search carried out for all the methods?

**Reasons To Accept:**

1. The method is simple, intuitive and logical. It builds upon a line of prior work on pseudo-labelling and self-training, which the authors adapt to LLMs.
1. The authors method achieves large improvements across a range of datasets, using fewer synthetic samples than prior methods.
1. Ablations show the importance of their iterative error refinement method, which generates

**Reasons To Reject:**

1. The theoretical analysis section is more of "theoretical intuition", as it makes several assumption which are the whole backbone of the author's method - such as "the error the small model makes on the small gold validation set can represent the distribution gap", "if an LLM is good enough it can perfectly extrapolate the errors", "theoretically, we can recover gold data distribution by simply combining the original seed data with additional data". All of these (and more), make this section completely redundant.
1. The "rationales" method is very similar to chain of thought style prompting.
1. Experiments are only done with one small model (distilbert) - perhaps other small models should also be considered to show the effectiveness for other models.

**Reproducibility:**

3: Could reproduce the results with some difficulty. The settings of parameters are underspecified or subjectively determined; the training/evaluation data are not widely available.

**Reviewer Confidence:**

4: Quite sure. I tried to check the important points carefully. It's unlikely, though conceivable, that I missed something that should affect my ratings.

---

> ### Author Rebuttal · Authors · 2023-08-29
>
> Dear Reviewer s5eU:
>
> Thanks a lot for your review, here is some further explaination of this paper:
>
> #### Answers to Reasons To Reject:
>
> ##### Response to the redundancy of the theoretical analysis section:
>
> We sincerely value the reviewer's perspective regarding our theoretical analysis section. While we acknowledge the merit in some of the concerns raised, our primary objective with this section was to offer an intuitive foundation for our methodologies, articulated through a mathematical lens. We did not intend to present these as unequivocal proofs, but rather as a means to provide readers with a deeper, high-level understanding of our approach and its underlying rationale. Given this intent, we believe that such a section enhances the comprehensiveness of our work rather than rendering it redundant. But as the reviewer points out, changing the name of this section can make the paper clearer, also, we will make this section more concise and may consider to move it to Appendix of the review consider it’s appropriate.
>
> ##### Response to similarity between rationals method and chain-of-thought:
>
> We recognize the comparison drawn between our "rationales" approach and the chain-of-thought style prompting as discussed in [1]. However, the core intent and execution of these methods are distinct. In our approach, the LLM is prompted to cultivate a comprehensive understanding of the overarching task set before it (as grounded by the rationale list). This acquired knowledge then informs subsequent data synthesis. In contrast, the chain-of-thought technique directs the LLM to tackle specific problems in a step-by-step progression.
>
> That’s to say, while our "rationales" method emphasizes a holistic grasp of the task, the chain-of-thought focuses on step-by-step problem-solving. We therefore believe they are different approaches. That being said, using chain-of-thought prompting for dataset synthesis is also not done before and is also a novel approach.
>
> ##### Response to experiments only done with one small model:
>
> We acknowledge the reviewer's concern regarding the restricted scope of our experiments by focusing solely on DistilBERT. To provide a more comprehensive evaluation of the S3 framework and to address this limitation, we have extended our experiments to encompass MiniLM [2], a model that shares a similar parameter count with DistilBERT but is better pre-trained. The following table presents the results of this additional experimentation, with the same dataset settings compared to our previous DistilBERT experiments:
>
> | Method/Dataset | IMDb | QNLI | RTE  | AdQA           |
> |----------------|------|------|------|----------------|
> |Gold Data       | 92.30| 91.00| 71.50| 22.97/36.59    |
> |ZeroGen         | 83.66| 70.11| 72.2 | 5.07/10.74     |
> |S3              | 89.55| 85.20| 76.17| 20.50/34.40    |
>
> These results clearly indicate that the S3 framework consistently outperforms ZeroGen across various datasets, irrespective of the model in use. Notably, similar to our DistilBERT findings, the S3 framework surpasses the performance of the gold data in the smaller RTE dataset.
>
> In light of these results, we hope the reviewer appreciates the broader efficacy of the S3 framework across different model architectures.
>
> #### Answers to Questions For Authors
>
> ##### Response to the R > 2 results (Question 1):
>
> We appreciate the reviewer's inquiry regarding the extension of our training to R>2 rounds. Indeed, we conducted experiments to explore this, specifically with the RTE dataset set for R=3. However, we observed that increasing the rounds of training beyond two does not lead to consistent improvement. Furthermore, when analyzing the results across different training rounds for the S3 method, the incremental benefits from the first to the second round of error extrapolation are quite marginal. Below are the detailed results to provide clarity:
>
> | Method/Dataset          | IMDb | QNLI | RTE  | AdQA           |
> |-------------------------|------|------|------|----------------|
> |SeedData                 | 86.86| 73.79| 59.93| 8.70/24.08     |
> |SeedData + 1 Round EES   | 88.47| 78.86| 71.12| 11.73/24.08    |
> |SeedData + 2 Rounds EES  | 89.00| 79.92| 73.29| 12.50/24.38    |
>
> We therefore we opted to a maximum of two rounds in our final experiments.
>
> ##### Response to difference of dataset size of GoldGen and S3 in QNLI dataset (Question 2):
>
> Sorry, that is a mistake we made during the paper writing, the actual dataset size for QNLI in GoldGen baseline is **$150k$** rather than $110k$. We have fixed this typo in the revised version; thanks a lot for pointing this out!
>
> ##### Response to parameter searching space (Question 3):
>
> We appreciate the reviewer's request for clarity on our hyper-parameter search. For your reference, the specific hyper-parameter search space we explored is outlined below:
>
> - Batch Size: 8, 16, 32
> - Learning Rate: 1E-5, 8E-6, 5E-6, 1E-6, 1E-7
> - Weight Decay: 1E-2, 1E-5, 1E-8
> - Epoch: 6, 8, 10, 15, 20
>
> To ensure a consistent and fair comparison, **we conduct the same hyperparameter search process for all the baselines as well as our approach**.
>
> #### Responses to other reviews:
>
> We deeply thank the reviewer's attentive feedback on typographical errors, grammar, and overall presentation. We have fixed them in the revised version.
>
> Regarding the concerns about reproducibility, we are committed to transparency and have therefore made our datasets available on an anonymous GitHub repository. You can access them at https://anonymous.4open.science/r/S3_Datasets-7421/README.md. We invite you to review the data at your convenience.
>
> [1] Chung, H. W., Hou, L., Longpre, S., Zoph, B., Tay, Y., Fedus, W., ... & Wei, J. (2022). Scaling instruction-finetuned language models. arXiv preprint arXiv:2210.11416.
> [2] Wang, W., Wei, F., Dong, L., Bao, H., Yang, N., & Zhou, M. (2020). Minilm: Deep self-attention distillation for task-agnostic compression of pre-trained transformers. Advances in Neural Information Processing Systems, 33, 5776-5788.

---

### Official Review · Reviewer_UzEm · 2023-08-12

**Soundness:** 4

**Excitement:**

3: Ambivalent: It has merits (e.g., it reports state-of-the-art results, the idea is nice), but there are key weaknesses (e.g., it describes incremental work), and it can significantly benefit from another round of revision. However, I won't object to accepting it if my co-reviewers champion it.

**Missing References:**

1. GeneratingTraining Data with Language Models: Towards Zero-Shot Language Understanding (https://arxiv.org/pdf/2202.04538.pdf) - SuperGen
2. SELF-GUIDED NOISE-FREE DATA GENERATION FOR EFFICIENT ZERO-SHOT LEARNING (https://arxiv.org/pdf/2205.12679.pdf) - SunGen

**Paper Topic And Main Contributions:**

The paper proposes a novel strategy to synthesize data that follows a distribution close to the gold dataset using LLMs by determining the rationale of a task and further using the error extrapolation synthesis to bring the distribution of the seed dataset closer to the gold dataset. From their experimental evaluation on several datasets, the proposed approach exhibits significant performance gains by smaller models as compared to previous synthetic data generation work. They also perform a theoretical analysis of the effectiveness of the proposed approach.

**Questions For The Authors:**

1. Won’t the model still perform better without the initial seed dataset? We directly start with evaluating the model on the eval gold set and then iteratively create samples similar to {x_mis, y_mis} and finally keep synthesizing the samples very close to the distribution of misclassified samples to improve performance.
2. Are there any checks for ensuring the quality of new samples generated? How different are the{x_add, y_mis} from the {x_mis, y_mis}?
3. It would be beneficial if there were a section about dataset diversity. How similar are the samples to each other? There has been a growing concern that human-not-in-loop data synthesis can lead to samples that are not an exact match but are very similar. For example: “I really enjoyed the new XXX movie” vs “I liked the XXX movie”.
4. Will there be an overlap in the new samples created owing to the constant buffer of rationales r_i that was queried from the LLM? Since we uniformly sample from the same rationale set, there could be significant overlap. Are there checks in place to ensure such scenarios do not occur?

**Reasons To Accept:**

1. Proposed a novel approach for synthetic data generation by identifying the rationale of the labels.
2. Proposed a novel error extrapolation-based synthesis to lower the distribution gap.
3. Thorough experimental evaluation has been conducted on real-world datasets along with ablation studies.

**Reasons To Reject:**

1. As shown in Algorithm 2 and Table 2, the paper suggests fine-tuning the small model, R (=2) number of times to extrapolate the error and reduce the dataset distribution gap for ~9% performance gain. This seems to be a costly approach in itself even though DistilBERT is being used.

**Reproducibility:**

5: Could easily reproduce the results.

**Reviewer Confidence:**

3: Pretty sure, but there's a chance I missed something. Although I have a good feel for this area in general, I did not carefully check the paper's details, e.g., the math, experimental design, or novelty.

---

> ### Author Rebuttal · Authors · 2023-08-29
>
> Dear Reviewer UzEm
>
> Thanks a lot for your review. Here are the answers to your question.
>
> #### Response to the High Cost of S3:
>
> Thank you for your feedback regarding the perceived computational cost of our method. You have highlighted concerns about the fine-tuning of DistilBERT multiple times, suggesting it might be resource-intensive. Let me address this concern with a detailed breakdown:
>
> We compare the floating-point operations (FLOPs) involved in fine-tuning the model with the S3 and ZeroGen synthesized datasets. For the DistilBERT model we used, based on OpenAI's report [1], the BERT family model costs $6$ FLOPs per training per token per parameter (i.e., $FLOP_{token, para} = 6$.). And based on [2], the DistilBERT has $numPara = 66 \times 10^6$ parameters. And the typical token length for one record is $ numToken_{rec} = 512$. Therefore, the training FLOP per record of data per epoch is:
>
> $
> FLOP_{rec} = FLOP_{token, para} * numToken_{rec} * numPara = 2.03 * 10^{11}
> $
>
> For ZeroGen, it typically uses $200k$ if data records and trains for an average $10$ epochs to achieve the best results based on our experiment. Thus, the fine-tuning cost for ZeroGen is:
>
> $
> FLOP_{ZeroGen} = FLOP_{rec} * 200k * 10 = 4.06 * 10^{17}
> $
>
> For our S3 method, in the first round of fine-tuning (using only seed data) dataset size is typically in $2/3$ size of final dataset (i.e. $51.2k$ records), and the second round of fine-tuning (using seed data + misclass generated data of first round) is in $5/6$ size of final dataset (i.e., $64.0k$), and the final round of fine-tuning (using seed data + 2 rounds of misclass generated data) have $76.8k$ data. And on average, our method needs $8$ epochs to achieve its best result. Therefore, the total FLOPs of fine-tuning DistilBERT for 3 times (2 for getting misclassified data, 1 for final fine-tuning) in our S3 frameworks are:
>
> $
> FLOP_{S3} = FLOP_{rec} * (51.2k + 64.0k + 76.8k) * 8 = 3.11 * 10^{17}
> $
>
> To conclude, we can see that our S3 methods use only 3/4 FLOPs compared to the ZeroGen baseline, even though we fine-tuned the model with multiple times.
>
>
> #### Answers to Questions For The Authors:
>
> ##### Response to the effect of the model without seed dataset (Question 1):
>
> Regarding the question, "Won't the model still perform better without the initial seed dataset...":
>
> We have indeed experimented with this approach, and unfortunately, the results were suboptimal. When we rely exclusively on training the smaller model without an initial seed dataset, it tends to overfit rapidly. We therefore conclude that the model needs the seed dataset to warmup. Otherwise it tends to significantly overfit the gold validation data.
>
>
> ##### Response to the quality check of $x_{add}$ generated by $x_{mis}$ (Question 2):
>
> Regarding your inquiry about the quality assurance of newly generated samples and their distinction from misclassified samples:
>
> We undertake a comprehensive post-hoc assessment to elucidate the distinctions between ${x_{add}, y_{mis}}$ and ${x_{mis}, y_{mis}}$. The objective is for the LLM-generated data to retain semantic similarity, while differing substantially in terms of word composition. Our analysis follows these steps:
>
> 1. **Sentence Encoding**:
>    For both misclassified data $D_{mis}$ and the additional generated data $D_{add}$, we employ DistilBERT to encode each $x$. This results in encoded sentences represented as $z_{mis}^{(i)}$ and $z_{add}^{(i)}$ respectively, where each encoded sentence resides in $\mathbb{R}^{d}$ (with $d = 768$ for DistilBERT).
> 2. **Cosine Similarity**:
>    By comparing the cosine similarity between $ z_{mis}^{(i)} $ and $ z_{add}^{(i)} $, we gauge semantic similarity. High cosine similarity indicates substantial semantic overlap.
> 3. **Edit Distance**:
>    To understand textual distinctiveness, we compute the edit distance between sentences $ x_{mis}^{(i)} $ and $ x_{add}^{(i)} $. If the edit distance approaches the sentence length, we infer the texts differ significantly in their composition.
>
> The results from our evaluation, split across two rounds of data generation, are tabulated below:
>
> | Rounds  | Label         | IMDb   | QNLI   | RTE    | AdQA   |
> | ------- | ------------- | ------ | ------ | ------ | ------ |
> | Round 1 | data num      | 3,291  | 28,532 | 883    | 26,605 |
> |         | cos sim       | 0.9445 | 0.9536 | 0.9367 | 0.9469 |
> |         | edit distance | 289.07 | 14.60  | 16.44  | 13.98  |
> |         | avg mis len   | 282.45 | 14.20  | 13.75  | 13.76  |
> |         | avg gen len   | 258.42 | 19.88  | 24.54  | 18.71  |
> | Round 2 | data num      | 2,882  | 22,568 | 639    | 24,927 |
> |         | cos sim       | 0.9557 | 0.9538 | 0.9397 | 0.9467 |
> |         | edit distance | 265.61 | 14.68  | 16.29  | 14.01  |
> |         | avg mis len   | 294.42 | 14.14  | 14.13  | 13.69  |
> |         | avg gen len   | 173.40 | 20.09  | 24.70  | 18.69  |
>
> The supplementary information on average misclassified data length (avg mis len) and average generated data length (avg gen len) provides context for the edit distances. This data unequivocally shows that while there's high semantic similarity (evidenced by the cosine similarity scores), the generated sentences are not mere copies of the misclassified samples. Thus, ensuring the quality of the newly generated data.
>
> ##### Response to check of dataset diversity (Question 3):
>
> We agree that dataset analysis is beneficial for dataset synthesis. However, there is no clear definition of "dataset complexity", which led us to exclude such an analysis. Nevertheless, we conduct analysis on the dataset diversity both quantitatively and qualitatively:
>
> **1. Quantitative Analysis:**
>
> For short synthesized sentences, as seen in QNLI, RTE, and AdQA datasets, we approached dataset analysis quantitatively. The brevity of these sentences means less dense information in the sentence encoding. Considering the high hidden dimension for encoding (e.g., 768 for DistilBERT), direct analysis can be inefficient due to the "curse of dimensionality". Hence, we used t-SNE [3] for dimension reduction:
>
> 1. Uniformally sample similar amount of data from gold data, S3 synthesized data, ZeroGen synthesized data. We have $ D_{gold} ' = \{(x_{gold}^{(i)}, y_{gold}^{(i)}) \}$ ($i$ from 1 to $n_1$), $D_{S3}' =  \{(x_{S3}^{(j)}, y_{S3}^{(j)}) \}$ ($j$ from $1$ to $n_2$), and $D_{ZeroGen}' =  \{(x_{ZeroGen}^{(k)}, y_{ZeroGen}^{(k)}) \}$ ($k$ from 1 to $n_3$), where $n_1, n_2, n_3$ should be similar.
>
> 2. Then encode the sentences by sentence encoder, we have $\{z_{gold}^{(i)} \}$, $\{z_{S3}^{(j)} \}$, $\{z_{ZeroGen}^{(k)} \}\subseteq \mathbb{R}^d$, where $d$ is the hidden state's dimension
>
> 3. Perform t-NSE on dataset $\{z_{gold}^{(i)} \} \cup \{z_{S3}^{(j)} \} \cup \{z_{ZeroGen}^{(k)} \}$ for all $i, j, k$ to reduce its dimension from $d$ to $2$. Name the processed data as $\{p_{gold}^{(i)} \}$, $\{p_{S3}^{(j)} \}$, $\{p_{ZeroGen}^{(k)} \} \subseteq \mathbb{R}^2$
>
> 4. Draw the reduced dimension points on a scattered plot to directly see the area of cover of our synthesized data. Because in the rebuttal section of openreview, we can not show the plots, you can check the plots of these datasets at this anonymous github https://anonymous.4open.science/r/S3_DiversityAnal-4AC4/README.md. Another way to show the plot's result is count how many points of $\{p_{gold}^{(i)}\}$ is in area of $A_{S3} = \cup_{j = 1}^{n_2} B_\gamma (p_{S3}^{(j)})$ and $A_{ZeroGen} = \cup_{j = 1}^{n_3} B_\gamma (p_{ZeroGen}^{(j)})$ where $B_\gamma (z)$ means the solid circle with center $z$ and radius $\gamma$. The results of the QNLI, RTE, and AdQA are as follows:
>
> | Dataset | S3 Coverage Rate | ZeroGen Coverage Rate |
> | ------- | ---------------- | --------------------- |
> | QNLI    | 76.35%           | 63.03%                |
> | RTE     | 73.95%           | 14.90%                |
> | AdQA    | 51.02%           | 46.00%                |
>
> This demonstrates the superior coverage and diversity of our S3 framework compared to ZeroGen.
>
> **2. Qualitative Analysis:**
>
> For longer text classification tasks, where sentence encoding results in dense information vectors that aren't amenable to t-SNE reduction, we conducted qualitative analysis. Here are a couple of diverse examples of generated data:
>
> - Example 1: "By chance, I stumbled upon this gem of a film and was completely entranced from beginning to end. As someone who appreciates good acting, I was blown away by the performances which felt so natural and unforced that I had to question whether it was scripted or improvised. The story itself was captivating and kept me invested throughout, leaving me wanting more even after the credits rolled. After a quick search on IMDb, I discovered that the writer-director has a strong track record with their previous film, Everyday People, which I am now eager to watch. It boggles my mind that a movie of this caliber can go unnoticed in the vast sea of blockbusters. This film deserves recognition and a wider audience. I implore everyone to seek it out and experience the magic for themselves."
> - Example 2: "While the cinematography in Mann's photographs of the Alberta Rocky Mountains is impressive, the rest of the film is sorely lacking. The acting, particularly from Jimmy Stewart and Walter Brennan, feels flat and uninspired. But what really ruins the entire experience is the absurd plot. A Mountie suggesting the people of Dawson City elect a marshal? And then gunfights breaking out on the streets? It's a laughable attempt to recreate the American Wild West in the Canadian North. Anyone with even a cursory knowledge of the Klondike gold rush will be shaking their head in disbelief at the ridiculousness of it all. Save yourself the time and skip this disaster of a film."
>
> These examples exhibit rich contexts and diverse patterns. For an exhaustive list, please refer https://anonymous.4open.science/r/S3_Datasets-7421/README.md.
>
> ##### Response to potential overlap problem of rational-based synthesized data (Question 4):
>
> The process we use for dataset synthesis with rationales is designed to minimize the potential for overlap. Let's break down the reasoning:
>
> 1. **Theoretical Analysis**: Each label's rationale set comprises 300 distinct rationales. For every dataset synthesis, we select 3 rationales from this set. This results in a total of $\binom{300}{3}$ (which equals 4,455,100) unique combinations of rationales. The vastness of this search space makes it improbable to generate a dataset with significant overlap.
>
> 2. **Randomness in Generation**: Our synthesis method uses a high temperature for generation, introducing a significant degree of randomness. This ensures that even if we use similar rationales, the generated outputs will differ considerably.
>
> Consider the following example where we used the exact same list of rationales: "low quality cinematography", "lack of attention to detail", and "too derivative of other movies or TV shows".
>
> - *Review 1*: "The movie review critiques the film's unoriginality and reliance on imitating existing content. The movie doesn't offer anything unique, drawing heavily from other cinema and TV shows."
>
> - *Review 2*: "A major flaw in the movie is its lack of innovation. The movie leans on existing plots and themes from previous content, resulting in a viewing experience that feels recycled and predictable."
>
> Both reviews convey similar sentiments, but the composition and phrasing are distinct. This illustrates our point: even with the same rationale, the generated data is diverse in both semantics and wording.
>
> #### Answers to low reproducibility:
> Since there are some concerns about our works' reproducibility, we decided to publish our synthesized dataset on anonymous git via https://anonymous.4open.science/r/S3_Datasets-7421/README.md.
>
>
> [1] Brown, T., Mann, B., Ryder, N., Subbiah, M., Kaplan, J. D., Dhariwal, P., ... \& Amodei, D. (2020). Language models are few-shot learners. Advances in neural information processing systems, 33, 1877-1901.
> [2] Sanh, V., Debut, L., Chaumond, J., \& Wolf, T. (2019). DistilBERT, a distilled version of BERT: smaller, faster, cheaper and lighter. arXiv preprint arXiv:1910.01108.
> [3] Van der Maaten, L., & Hinton, G. (2008). Visualizing data using t-SNE. Journal of machine learning research, 9(11).

---

### Meta-Review · Area_Chair_1KBm · 2023-09-18

**Recommendation:** 3

**Metareview:**

A simple and effective approach to do data generation. Written well and easy to understand. It's technically sound and show good performance. However, for originality, it's not a very novel idea that would surprise people.

Pros:

1. A simple and effective method to do data generation.

2. Great performance on some benchmark datasets.

Cons:

1. It's not a particularly interesting method. None reviewer showed the excitement high enough to support the paper. It reads like reviewers all feel it's reasonable to get this performance and there is no analysis showing why pervious(or simple heuristic) methods would fail and why the problem is extremely difficult.

---

### Decision · Program_Chairs · 2023-10-07

**Decision:**

Accept-Findings

**Comment:**

A simple and effective approach to do data generation. Written well and easy to understand. It's technically sound and show good performance. However, for originality, it's not a very novel idea that would surprise people.

Pros:

1. A simple and effective method to do data generation.

2. Great performance on some benchmark datasets.

Cons:

1. It's not a particularly interesting method. None reviewer showed the excitement high enough to support the paper. It reads like reviewers all feel it's reasonable to get this performance and there is no analysis showing why pervious(or simple heuristic) methods would fail and why the problem is extremely difficult.